# An Extraction Method for Large Gradient Three-Dimensional Displacements of Mining Areas Using Single-Track InSAR, Boltzmann Function, and Subsidence Characteristics

Kegui Jiang [1], Keming Yang [1,*], Yanhai Zhang [2], Yaxing Li [1], Tingting Li [1] and Xiangtong Zhao [1]

1 College of Geoscience and Surveying Engineering, China University of Mining and Technology—Beijing, Beijing 100083, China; jkg@student.cumtb.edu.cn (K.J.)
2 General Defense Geological Survey Department, Huaibei Mining Co., Ltd., Huaibei 235000, China
* Correspondence: ykm@cumtb.edu.cn

**Abstract:** This paper presents an extraction method for large gradient three-dimensional (3-D) displacements of mining areas using single-track interferometric synthetic aperture radar (InSAR), Boltzmann function, and subsidence characteristics. This is mainly aimed at overcoming the limitations of surface deformation monitoring in mining areas by using single-track InSAR technology. One is that the rapid and large gradient deformation of the mine surface usually leads to image decoherence, which makes it difficult to obtain correct deformation information. Second, the surface deformation monitored by InSAR is only one-dimensional line of sight (LOS) displacement, and thus it is difficult to reflect the real 3-D displacements of the surface. Firstly, the Boltzmann function prediction model (BPM) is introduced to assist InSAR phase unwrapping; thus the missing large gradient deformation phase of InSAR is recovered. Then, the subsidence characteristics in mining horizontal (or near-horizontal) coal seams are used as prior knowledge for theoretical derivation, and a 3-D displacement extraction model of coal seam mining with single-track InSAR is constructed. The feasibility of the method is verified by simulating LOS displacements with random noise and underestimation phenomenon caused by the large gradient deformation as InSAR observations. The results show that the root mean square error (RMSE) of 3-D displacements on the observation line calculated by the proposed method is 21.5 mm, 19.0 mm, and 32.9 mm, respectively. Based on the single-track Sentinel-1 images, the method in this paper was applied to the extraction of surface 3-D displacements in the Huaibei coal mine, and the experimental results show that the extracted 3-D displacements are in good agreement with that of measurement by the surface observation station. The proposed method can adapt to limited InSAR acquisitions and complex monitoring environments.

**Keywords:** three-dimensional displacements; large gradient deformation; InSAR; mining area; Boltzmann function; subsidence characteristics

## 1. Introduction

As an important energy resource base, coal is strong support for industrial modernization and social development in China. However, with the increasing energy demand, while coal mining has created huge value for the domestic economy, the disasters caused by it cannot be ignored: the subsidence and collapse caused by large-scale continuous mining damage the ground infrastructure, induce various geological disasters, and pose a huge threat to the safety of people and property in mining areas [1,2]. Therefore, monitoring and predicting the three-dimensional (3-D) displacements of the mine surface plays a vital role in assessing potential geological disasters, ensuring safe production, and analyzing the settlement mechanism of the mining area [3]. Traditional leveling and GPS measurement technologies are most widely used in surface deformation monitoring in mining areas. Although such methods have high monitoring accuracy, they have disadvantages such as small range, heavy workload, high cost, low efficiency, and easily damaged measuring

points. In addition, the spatial-temporal resolution of "linear" deformation observation is low, and it is difficult to completely reflect the spatial-temporal evolution of the surface 3-D deformations of the entire mining area; thus hindering the comprehensive monitoring of mining subsidence and the development of related theoretical research.

Interferometric synthetic aperture radar (InSAR) is a new type of active surface deformation monitoring technology, which has the advantages of all-weather, all-time, low cost, large coverage, and high spatial resolution. InSAR technology provides a new method for monitoring and predicting displacement in mining areas [4–6]. However, InSAR technology has inherent limitations in monitoring surface displacements in mining areas. First, when the surface deformation gradient caused by mining exceeds the detection capability of synthetic aperture radar (SAR) satellites, the SAR images at the target will be decoherent, which will easily lead to the failure of interferometric phase unwrapping or the underestimation of unwrapping results. Second, due to the characteristics of SAR sensor squint imaging, the surface deformations monitored by InSAR are only one-dimensional displacement, while the mining-induced surface deformations are actually 3-D displacements along the vertical and horizontal directions. How to adapt to the InSAR monitoring characteristics to extract the 3-D displacements of the whole basin is a difficult point that plagues the application of this technology to monitoring displacement in mining areas.

At present, to overcome the above difficulties, the following methods have been developed by scholars: Method (I) multi-track InSAR [7–10], InSAR + Offset Tracking/MAI [11–14], and InSAR + GPS [15–17]; Method (II) Prior model + InSAR [18–20]; Method (III) InSAR + probability integral method (PIM) [21–24]. Method (I) can theoretically obtain one-dimensional or two-dimensional observations in multiple directions, thereby the number of additional observation equations is increased and the 3-D deformations observation information is supplemented. Furthermore, the mining-induced large gradient deformation can be captured by Offset Tracking to a certain extent. However, in practical application, this type of method is limited by the lack of observation data, the low resolution of the image, and the accumulation of water in the subsidence area. Based on Method (II), additional constraints are constructed through the prior relationship of 3-D displacements, and then the extraction of surface 3-D displacements based on InSAR observations is realized. Compared with Method (I), Method (II) is less affected by the observation data, and the accuracy of estimated displacement in the north–south direction is improved. The disadvantage is that it is greatly affected by the reliability of the prior model, and the extraction of large gradient 3-D deformations of the mining coal seam cannot be applied. For Method (III), given the difficulty of obtaining the rapid and large gradient deformation in the center of the subsidence basin with InSAR technology, the main idea of this method is first to establish a relationship model between the surface displacement along the line of sight (LOS) direction and the surface 3-D displacements, and inversion of deformation prediction parameters based on nonlinear theory is implemented. Thus, 3-D displacements of the whole basin can be obtained through the prediction of the model. The decoherence problem of monitoring large gradient deformation by InSAR technology in mining areas (defects of Methods (I) and (II)) is circumvented by Method (III) to a certain extent. However, the solved deformation of the PIM with faster boundary convergence is quite different from the measured value, and in consequence, the model parameters inverted by the PIM are not consistent with the reality (especially as the subsidence coefficient is smaller than the real value), which leads to a smaller value of the predicted large gradient deformation of the subsidence basin.

Aiming at the above problems, an extraction method for large gradient three-dimensional displacements of mining areas using single-track InSAR, Boltzmann function, and subsidence characteristics is proposed in this paper. With the Boltzmann function prediction model (BPM) as an auxiliary condition, the phase of the predicted displacement is used to assist interferometric phase unwrapping in mining areas; hence, the large gradient deformation of the surface along the LOS direction is recovered. In addition, according to the subsidence symmetry characteristics of the horizontal or gently inclined working

face of mining coal, combined with the geometric projection relationship between the LOS displacement monitored by InSAR and the real 3-D displacements of the surface, the extraction model for the surface 3-D displacements of mining coal is constructed.

The rest of this paper is structured as follows. Section 2 describes the methodology for extracting large gradient 3-D deformation. The validations of the proposed method by simulated and real data experiments are carried out in Sections 3 and 4, followed by the discussion in Section 5. Section 6 presents the conclusion.

## 2. Methodology

To overcome the difficulty of unwrapping a large gradient deformation phase monitored by InSAR, the BPM (compared with the traditional PIM, the model has a better fit at the boundary of the deformation basin [25,26]) is introduced to assist the phase unwrapping of InSAR. Furthermore, according to surface subsidence characteristics of the mining rectangular working face, the extraction model for 3-D displacements with InSAR technology is constructed by combining with the geometric projection relationship. Finally, the extraction method for large gradient 3-D displacements of mining areas based on single-track InSAR is constructed by summarizing the above two processes.

### 2.1. BPM-Assisted Phase Unwrapping Model for InSAR
### 2.1.1. Boltzmann Function Prediction Model

The Maxwell distribution was promoted by Boltzmann to the Maxwell–Boltzmann distribution, and this research result has been widely used [27–29]. The equation of the Boltzmann mathematical model is $y = \frac{A_1 - A_2}{1 + e^{(x-x_0)/B}} + A_2$; thus, it is easy to find that the curve shape of this function is S-shaped, which is similar to the subsidence curve shape of the main section of the surface deformation basin during semi-infinite mining. The subsidence prediction of the PIM model is shown as:

$$W(x) = \frac{W_0}{2}\left[\mathrm{erf}\left(\frac{\sqrt{\pi}}{r}x\right) + 1\right] \tag{1}$$

In Equation (1), $W_0 = m \cdot q \cdot \cos\alpha$ represents maximum subsidence, as shown in Figure 1, where $m$, $q$, and $\alpha$ are the mining height, the subsidence coefficient, and the inclined angle of the working face, respectively. $r = H/\tan\beta$ represents the main influence radius, where $H$ is the mining depth and $\beta$ is the angle of major influence. $x$ is the position of the point on the main section. $\mathrm{erf}(x) = \frac{2}{\sqrt{\pi}}\int_0^x \exp(-u^2)du$ is the probability density function. Combining the form of the Boltzmann function and Equation (1), the subsidence prediction of the main section of the surface deformation basin during semi-infinite mining based on the Boltzmann function is determined as:

$$W(x) = \frac{W_0}{1 + e^{-x/R}} \tag{2}$$

where $R = r/4.13$ denotes a new main influence radius. By differentiating the formula, the unit influence function of the BPM can be obtained:

$$w_e(x) = dW(x) = \frac{\exp\left(\frac{-x}{R}\right)}{R\left(1 + \exp\left(\frac{-x}{R}\right)\right)^2} \tag{3}$$

According to the basic assumption of stochastic medium theory, it is considered that the following two conditions are satisfied after the mining of infinitely small coal seam units: (1) the overall volume of the overlying strata remains unchanged; (2) the movement and deformation of the overlying strata are distributed continuously. Hence, when deriving the horizontal displacement of the surface caused by the mining of infinitely small coal

seam units, the derivation experience of the PIM and elastic mechanics knowledge is used for reference, and the unit horizontal displacement form is determined:

$$U_e(x) = B\frac{dw_e(x)}{dx} \tag{4}$$

where $B$ is a constant, which can be integrated to obtain the horizontal displacement $U(x)$ of any ground point along the $x$ direction, as follows:

$$U(x) = W_0\frac{B}{R}\frac{\exp\left(\frac{-x}{R}\right)}{\left(1 + \exp\left(\frac{-x}{R}\right)\right)^2} = b'W_0\frac{\exp\left(\frac{-x}{R}\right)}{\left(1 + \exp\left(\frac{-x}{R}\right)\right)^2} \tag{5}$$

where $b' = B/R = 4.13 \cdot b$ and $b$ is the horizontal movement coefficient.

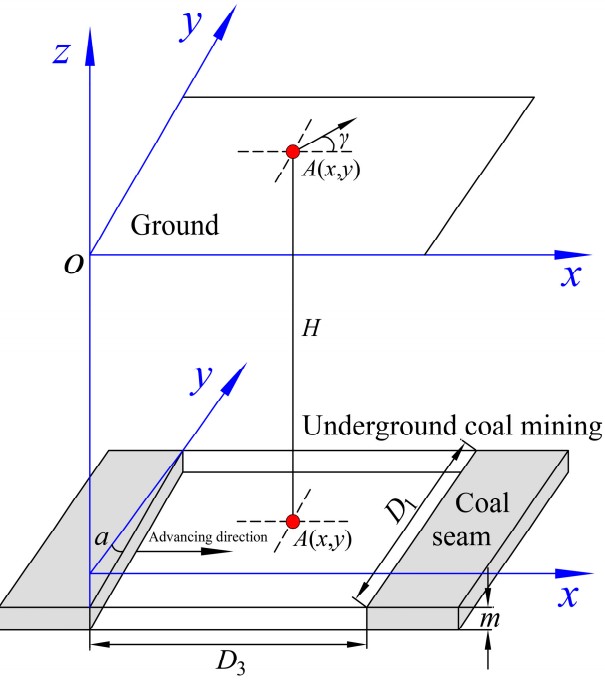

**Figure 1.** Principle for predicting the deformation of surface point $A(x,y)$ caused by underground coal mining.

The calculation formulas for the surface displacements along the vertical and horizontal direction by the finite mining of the strike main section can be derived as:

$$\begin{cases} W^o(x) = W(x) - W(x - l) \\ U^o(x) = U(x) - U(x - l) \end{cases} \tag{6}$$

Then, the calculation formulas for the surface displacements along the vertical and horizontal direction by the finite mining of the inclination main section is:

$$\begin{cases} W^o(y) = W(y) - W(y - L) \\ U^o(y) = U(y) - U(y - L) + W^o(y)\cot\theta \end{cases} \tag{7}$$

In Equations (6) and (7), $l = D_3 - S_3 - S_4$ and $L = D_1 - S_1 - S_2$ is the calculation length of the working face along the strike and inclination direction. Furthermore, $S_1$, $S_2$, $S_3$, and $S_4$ denote the offsets of the inflection points in the lower ribside, the upper ribside, the start production line, and the stop production line.

Based on the Boltzmann function model, the surface displacements along the vertical and horizontal direction of any point $A(x,y)$ caused by underground mining are, respectively:

$$\begin{cases} W(x,y) = \frac{1}{W_0}W^o(x)W^o(y) \\ U(x,y,\gamma) = \frac{1}{W_0}[U^o(x)W^o(y)cos\gamma + U^o(y)W^o(x)sin\gamma] \end{cases} \tag{8}$$

where $\gamma$ is the anticlockwise angle between the prediction direction and the positive direction of the $x$-axis.

In summary, the model parameters of the BPM can be divided into two types of parameter systems, namely, the geological mining parameter systems $G = [m, \alpha, \varphi, H, D_3, D_1]$, where $\varphi$ is the azimuth of working face along the strike direction, which directly participates in the conversion of the coordinate system, and the prediction parameter systems $P = [q, \tan\beta, b, \theta, S_1, S_2, S_3, S_4]$. The geological mining parameter systems $G$ can be obtained from the collected geological mining data of the mining area. In addition, the prediction parameter systems $P$ is consistent with the model parameters of the PIM, and the empirical values of the prediction parameters in the mining area or adjacent mining areas can be used.

### 2.1.2. InSAR Phase Unwrapping Model

Phase unwrapping is the process of recovering the real phase from the phase wrapped between $(-\pi, \pi]$; therefore, the accuracy of the result will directly affect the quality of the final deformation product [30–32]. Phase gradient estimation is an important factor affecting the accuracy of unwrapping results. However, the accuracy of phase gradient estimation is not ideal in the mining area with high noise and rapid and large gradient changes, which leads to underestimation of unwrapping results and even unwrapping failure [20,33]. Aiming at the above problems, the BPM-assisted InSAR phase unwrapping model is constructed based on Section 2.1.1.

The 3-D geometric decomposition principle of the LOS displacement monitored by InSAR is shown in Figure 2, where $\delta$ represents the flight azimuth of the satellite, which is the angle between the flight direction of the satellite and true north, and $\xi$ is the incident angle of the radar signal. PL represents the projection line of the $DLOS$ on the target position on the horizontal plane; $U_E$, $U_N$, and $W$ are the displacements along the east direction, north direction, and vertical direction, respectively, and represent real 3-D displacements of the target position on the ground; $U_{E, PL}$ represents the projection of $U_E$ on PL; $U_{N, PL}$ represents the projection of $U_N$ on PL; $U_{PL}$ represents the vector sum of $U_{E, PL}$ and $U_{N, PL}$ on the PL line; $U_{PL, LOS}$ represents the projection of $U_{PL}$ on the LOS direction; $W_{LOS}$ represents the projection of $W$ on the LOS direction; that is, $DLOS$ is the vector sum of $U_{PL, LOS}$ and $W_{LOS}$. The cumulative LOS displacement of the pixel $j$ where the surface point $A(x, y)$ is located in the mining working face is:

$$DLOS^j = W^j \cos\xi - U_E^j \sin\xi \cos\delta + U_N^j \sin\xi \sin\delta \tag{9}$$

According to the principle of InSAR interferometric, the relationship between the monitored LOS displacement and the monitored phase of pixel j can be expressed as:

$$\phi_{mon}^j = -\frac{4\pi}{\lambda}DLOS_{mon}^j \tag{10}$$

Based on the relationship between the wrapped phase and the unwrapped phase, we can obtain:

$$\phi_{mon}^j = \varphi_{mon}^j + k_{mon}^j \times 2\pi \tag{11}$$

In Equations (10) and (11), $\lambda$ is the wavelength of the SAR signal; $\varphi_{mon}^j$ is the wrapped phase of InSAR interferometric; $k_{mon}^j$ is the integer cycle ambiguity of the InSAR phase. In general, due to the complex monitoring conditions of the mining area, the closer to the

center of the subsidence basin, the smaller the monitored value of $k_{mon}^j$ is than the real integer cycle ambiguity $k_{real}^j$. Therefore, it is necessary to adopt a certain method to restore the real integer cycle ambiguity.

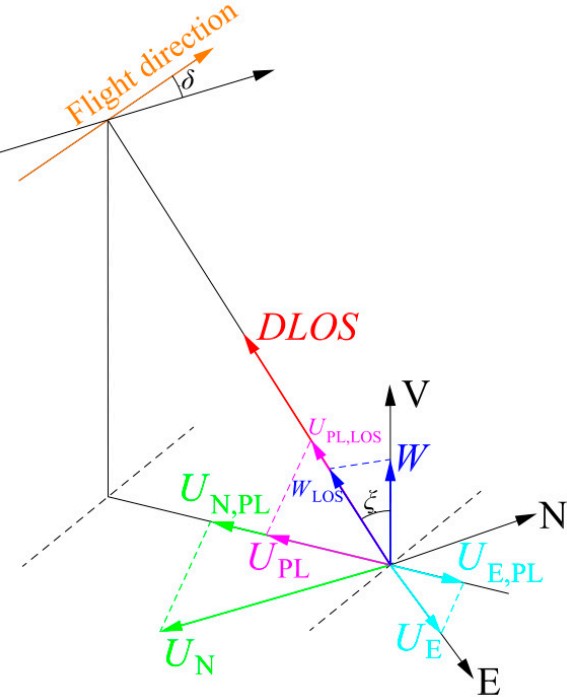

**Figure 2.** Geometric projection relationship between the LOS displacement (*DLOS*) monitored by InSAR and the real 3-D displacements of the surface.

Furthermore, using the geological mining parameters and the prediction parameters, the 3-D displacements ($W^j$, $U_E^j$, and $U_N^j$) at the corresponding pixel of the target area is predicted based on the BPM. Then, combined with the geometric parameters of the satellite, the synthesis of predicted LOS displacement $DLOS_{pre}^j$ in the target area is carried out by using Equation (9). Thus, the predicted unwrapped phase $\phi_{pre}^j$ of displacements can be derived from predicted LOS displacement as shown in Equation (12), and the relationship between the predicted unwrapped phase $\phi_{pre}^j$ and the predicted wrapped phase $\varphi_{pre}^j$ is shown in Equation (13):

$$\phi_{pre}^j = -\frac{4\pi}{\lambda} DLOS_{pre}^j \tag{12}$$

$$\phi_{pre}^j = \varphi_{pre}^j + k_{pre}^j \times 2\pi \tag{13}$$

where $k_{pre}^j$ is the integer cycle ambiguity of the predicted phase.

Next, the residual phase of the monitored phase is obtained by removing the predicted phase from the monitored phase, as shown in Equation (14). The method of unwrapping the residual phase is adopted to reduce the difficulty of monitoring phase unwrapping, as shown in Equation (15):

$$\phi_{res}^j = \phi_{mon}^j - \phi_{pre}^j = \left( \varphi_{mon}^j - \varphi_{pre}^j \right) + \left( k_{mon}^j - k_{pre}^j \right) \times 2\pi \tag{14}$$

$$\phi_{res}^j = \varphi_{res}^j + \left( k_{mon}^j - k_{pre}^j + m^j \right) \times 2\pi \tag{15}$$

where $\phi_{res}^{j}$ is the residual phase; $\varphi_{res}^{j} = mod\left[\left(\varphi_{mon}^{j} - \varphi_{pre}^{j}\right) + \pi, 2\pi\right] - \pi$ denotes the wrapped phase of $\phi_{res}^{j}$, and $\left(k_{mon}^{j} - k_{pre}^{j} + m^{j}\right)$ is the integer cycle ambiguity of the residual phase.

After the above operation, assuming that the solved residual phase after unwrapping is $\phi_{res,sol}^{j}$, according to Equation (14), the solved displacement phase can be expressed as:

$$\phi_{sol}^{j} = \phi_{pre}^{j} + \phi_{res,sol}^{j} \tag{16}$$

Finally, according to the principle of InSAR technology, it can be known that the solved displacement phase of surface point $A(x, y)$ corresponding to pixel $j$ can be converted into the LOS displacement by using Equation (10).

It needs to be further explained that the decoherence in some areas (water body, high noise, and large gradient deformation on the ground cause the failure of unwrapping some pixels) monitored by InSAR leads to the loss of LOS displacement information of the corresponding pixels; hence, before carrying out the above operations, the missing pixels need to be interpolated. In addition, the lack of a large number of pixels in the center of the deformation basin is likely to cause the LOS displacement after interpolation to be much smaller than the real value, and we will discuss the extraction of surface 3-D displacements under this condition in simulated experiments.

### 2.2. 3-D Displacements Extraction Model Based on Symmetrical Characteristics of Mining Subsidence

#### 2.2.1. Surface Subsidence Symmetrical Characteristics of Coal Seam Mining

After the underground coal seam is mined, the movement and deformation of the surface and its spatial distribution have certain regularity [34,35]. For the longwall mining method with full mechanization, after the surface deformation caused by mining horizontal or near-horizontal rectangular working face is stabilized, since the inclined angle of the coal seam is close to zero, the geometric characteristics and physical properties of overlying strata and surface are all isotropic, and the displacement of overlying rocks and surface caused by mining present a symmetrical distribution. Further analysis shows that the axis of symmetry is the strike main section and inclination main section passing through the center of the goaf and parallel (or perpendicular) to the advancing direction of the working face, that is, the approximately symmetrical axis of the surface subsidence and horizontal movement along the strike direction are the inclination main section. Similarly, the approximately symmetrical axis of the surface subsidence and horizontal movement along the inclination direction is the strike main section. Using these relationships as prior knowledge for deriving theoretical models, an extraction model for 3-D displacements based on the symmetrical characteristics of mining subsidence is constructed.

As shown in Figure 3, *P*1 (*x*1, *y*1), *P*2 (*x*2, *y*2), *P*3 (*x*3, *y*3), and *P*4 (*x*4, *y*4) are the points in each quadrant where the ground surface above the working face is symmetrical about the strike and inclination main section. Based on prior knowledge of symmetrical characteristics of mining subsidence, the vertical displacement of *P*1, *P*2, *P*3, and *P*4 are equal in magnitude and the same direction. The strike displacement of *P*1 and *P*2 are equal in magnitude and the same direction, and the inclination displacement of *P*1 and *P*2 is equal in magnitude and the opposite direction. The strike displacement of *P*1 and *P*3 are equal in magnitude and the opposite direction, and the inclination displacement of *P*1 and *P*3 is equal in magnitude and the opposite direction. The strike displacement of *P*1 and *P*4 are equal in magnitude and the opposite direction, and the inclination displacement of *P*1 and *P*4 is equal in magnitude and the same direction.

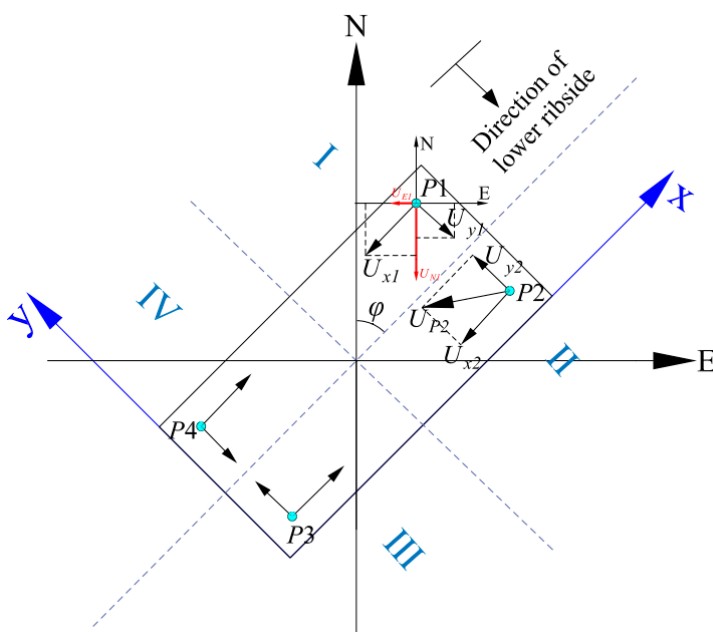

**Figure 3.** Vector relationship of the horizontal displacements of the symmetrical point on the ground.

2.2.2. 3-D Displacements Extraction Model of Coal Seam Mining with Single-Track InSAR

The north and east displacements of point $P1$ can be obtained by projecting the strike and inclination displacements of point $P1$ in the north and east directions, respectively, as follows:

$$\begin{cases} U_{N1} = -U_{x1} \cos \varphi - U_{y1} \sin \varphi \\ U_{E1} = -U_{x1} \sin \varphi + U_{y1} \cos \varphi \end{cases} \tag{17}$$

where $\varphi$ is the azimuth of the working face along the strike direction, and its value range is 0–2$\pi$; $U_{x1}$ and $U_{y1}$ are the displacement components of point $P1$ along the strike and inclination direction, respectively, and the representation of the displacement components of point $P2$, $P3$, and $P4$ along the strike and inclination direction is the same as that of point $P1$.

Similarly, the displacement components of points $P2$, $P3$, and $P4$ in the second, third, and fourth quadrants, respectively, along the north direction and east direction can be obtained:

$$\begin{cases} U_{N2} = -U_{x2} \cos \varphi + U_{y2} \sin \varphi \\ U_{E2} = -U_{x2} \sin \varphi - U_{y2} \cos \varphi \\ U_{N3} = U_{x3} \cos \varphi + U_{y3} \sin \varphi \\ U_{E3} = U_{x3} \sin \varphi - U_{y3} \cos \varphi \\ U_{N4} = U_{x4} \cos \varphi - U_{y4} \sin \varphi \\ U_{E4} = U_{x4} \sin \varphi + U_{y4} \cos \varphi \end{cases} \tag{18}$$

Substituting Equations (17) and (18) into Equation (9) can get:

$$\begin{bmatrix} DLOS_1 \\ DLOS_2 \\ DLOS_3 \\ DLOS_4 \end{bmatrix} = \begin{bmatrix} PM1 \times |\overrightarrow{W_1}| + PM2 \times |\overrightarrow{U_{x1}}| - PM3 \times |\overrightarrow{U_{y1}}| \\ PM1 \times |\overrightarrow{W_2}| + PM2 \times |\overrightarrow{U_{x2}}| + PM3 \times |\overrightarrow{U_{y2}}| \\ PM1 \times |\overrightarrow{W_3}| - PM2 \times |\overrightarrow{U_{x3}}| + PM3 \times |\overrightarrow{U_{y3}}| \\ PM1 \times |\overrightarrow{W_4}| - PM2 \times |\overrightarrow{U_{x4}}| - PM3 \times |\overrightarrow{U_{y4}}| \end{bmatrix} \tag{19}$$

with:

$$\begin{cases} PM1 = \cos \xi \\ PM2 = \sin \xi \cos \delta \sin \varphi - \sin \xi \sin \delta \cos \varphi \\ PM3 = \sin \xi \cos \delta \cos \varphi + \sin \xi \sin \delta \sin \varphi \end{cases} \tag{20}$$

where $DLOS_1$, $DLOS_2$, $DLOS_3$, and $DLOS_4$ represent the LOS displacement monitored by InSAR at $P1$, $P2$, $P3$, and $P4$.

According to the symmetrical characteristics of surface subsidence in the mining of horizontal or near-horizontal rectangular working face, the relationship between symmetrical points in each quadrant is constructed, as follows:

$$
\begin{cases}
U_x = |\overrightarrow{U_{x1}}| = |\overrightarrow{U_{x2}}| = |\overrightarrow{U_{x3}}| = |\overrightarrow{U_{x4}}| \\
U_y = |\overrightarrow{U_{y1}}| = |\overrightarrow{U_{y2}}| = |\overrightarrow{U_{y3}}| = |\overrightarrow{U_{y4}}| \\
W = |\overrightarrow{W_1}| = |\overrightarrow{W_2}| = |\overrightarrow{W_3}| = |\overrightarrow{W_4}|
\end{cases}
\tag{21}
$$

Substituting Equation (21) into Equation (19) can get:

$$
\begin{bmatrix} DLOS_1 \\ DLOS_2 \\ DLOS_3 \\ DLOS_4 \end{bmatrix}
=
\begin{bmatrix}
PM1 & PM2 & -PM3 \\
PM1 & PM2 & PM3 \\
PM1 & -PM2 & PM3 \\
PM1 & -PM2 & -PM3
\end{bmatrix}
\begin{bmatrix} W \\ U_x \\ U_y \end{bmatrix}
\tag{22}
$$

Using the principle of least squares, $W$, $U_x$, and $U_y$ can be calculated, and the $U_E$ and $U_N$ can also be calculated through the projection relationship (Equations (17) and (18)). Based on the above process, 3-D displacements of any point on the surface can be extracted.

Compared with the Method (II) described in the introduction, the 3-D displacements extraction model with single-track InSAR constructed in this section does not need to collect prior model parameters $b$, $H$, and $\tan\beta$ in advance, and the influence of model parameter errors on the calculation of the 3-D displacement fields is avoided.

### 2.3. Construction Process of Method

The specific process of the extraction method for large gradient 3-D deformation of mining areas based on single-track InSAR is shown in Figure 4.

(1) InSAR process: The multiple single-track SAR images covering the mining area is collected, and then the InSAR interferometric processing is performed for the whole mining period; hence, the cumulative LOS displacement phase of InSAR is solved. The measured interferometric wrapped phase and the integer cycle ambiguity of the whole mining period are obtained through the phase reverse wrapping.

(2) Prediction process: First, geological mining materials and the experience value of prediction parameters in the mining area or adjacent mining area are obtained. Then, based on the BPM, the vertical displacement, the north displacement, and the east displacement are predicted, respectively, and the predicted LOS displacement is synthesized by using Equation (9). Finally, the operations of displacement into phase and phase reverse wrapping are executed, thus, the predicted wrapped phase and the integer cycle ambiguity of the whole mining period are calculated.

(3) Calculation of LOS displacement: By taking the difference between the measurement phase and prediction phase in the first two steps, the wrapped phase of the residual phase in the differential interference diagram is solved. Moreover, the unwrapped residual phase is calculated by the unwrapping of the residual phase. Based on superimposing the unwrapped residual phase and predicted phase of LOS displacement, the final solution phase of LOS displacement is obtained. Finally, the LOS displacement can be solved by using Equation (12).

(4) Extraction of 3-D displacements: Based on the solved LOS displacement field, the coordinates and LOS displacement values in the first quadrant of the coordinate system are read pixel by pixel. Furthermore, the coordinates and LOS displacement values of corresponding pixels in other quadrants can be found according to the symmetrical relationship. Thus, based on Equation (22), the vertical displacement, the north displacement, and the east displacement can be solved by using the least squares method. Until all the pixels in the quadrant are calculated, the surface 3-D displacement field is output.

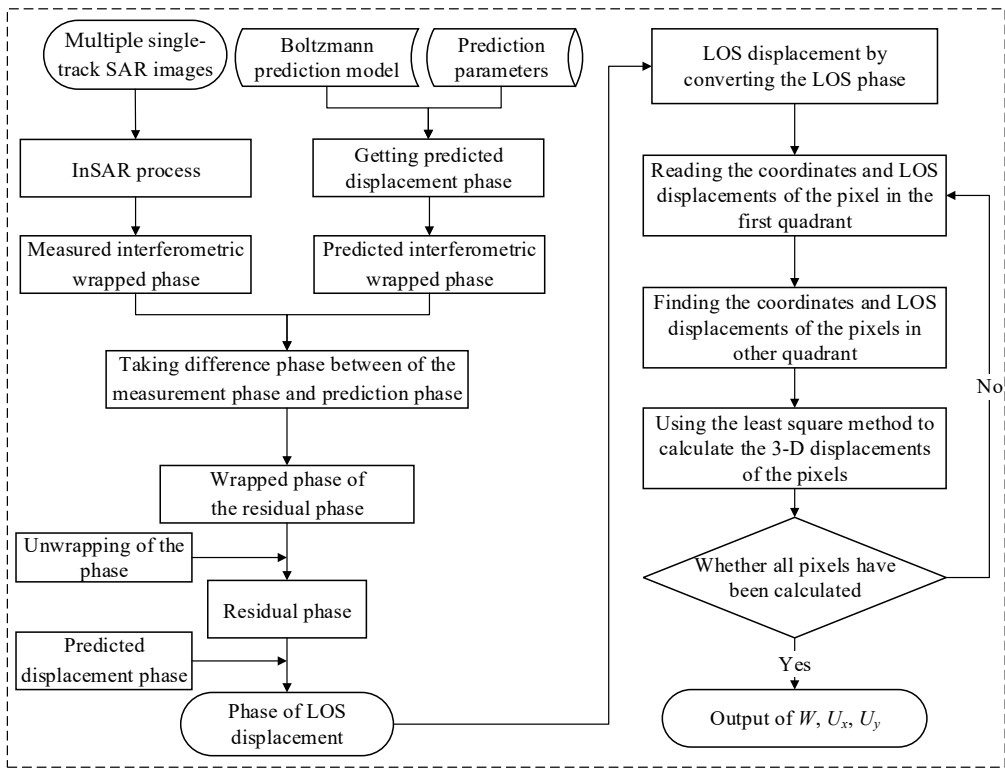

**Figure 4.** Technology roadmap of the construction method.

## 3. Simulated Experiments

### 3.1. Overview of Simulated Geological Mining Conditions and Satellite Data

Taking the coal measure strata conditions in the Huaibei mining area as the background, the geological mining conditions of the simulated working face are as follows: the lithology of the overlying strata is moderately hard; a mining size of the working face is $D_3 \times D_1 = 600 \times 300$ m; an average mining height of the coal seam is $m = 3$ m; an inclined angle is $a = 0°$; a mining depth is $H = 400$ m; an azimuth angle of strike direction is $\varphi = 45°$. The roof management method is the longwall caving method. The simulated working face is advanced along the strike, with mining the full length along the inclination at one time. Engineering practice shows that the relationship between underground mining and surface response in the Huaibei mining area conforms to the BPM, and simulated model prediction parameters are $q = 0.8$, $\tan\beta = 2.0$, $b = 0.3$, $\theta = 87°$, $S_1 = S_2 = S_3 = S_4 = 0$ m. Simulated satellite data refers to the shooting parameters of the Sentinel-1 satellite in the Huaibei area, with a simulated azimuth angle of flight direction $\delta = 347.2°$ and a LOS incident angle $\xi = 37.5°$.

During simulation of the LOS displacement by InSAR monitoring, first, the grid on the ground above the working face is divided along the east–west direction and north–south direction, and satellite image pixels are simulated by using the grid. Then, according to the BPM and the simulated prediction parameters, the cumulative 3-D displacements on the ground caused by underground mining (that is, vertical displacement, north displacement, and east displacement, respectively, as shown in Figure 5a–c) are predicted. In addition, based on Equation (9), the LOS displacement value of any target pixel is synthesized, as shown in Figure 5d. In addition, when monitoring the deformation of coal mining, we will analyze the phase noise and anomalies that may be introduced by InSAR images from two aspects. First, due to SAR images usually containing random noise and atmospheric delay, simulated additive noise is added to the LOS deformation, and the simulated additive noise in this experiment is Gaussian noise with a mean of 10 and a standard deviation of 50 [18,36]. The simulated noise is shown in Figure 5e. Second, because the surface deformation caused by mining usually has a large gradient and rapid characteristics, which brings serious noise and dense interference fringes to InSAR monitoring, the surface deformation information

obtained by the unwrapping process of the traditional InSAR is often missing or seriously underestimated. Hence, the above situations lead to the monitored LOS displacement being much smaller than the real value. To explore the influence of these conditions on the extraction of surface 3-D displacements, we assume that the InSAR method cannot obtain all the surface deformation information caused by the mining of the working face. By analyzing the SAR satellite monitoring capability, time and space resolution, and the advancing speed of the working face, the maximum cumulative deformation of InSAR monitoring is set at 800 mm [21,37]. To sum up, by introducing simulated Gaussian noise and setting the maximum monitoring value, the simulated LOS deformation with noise and limited maximum observation is shown in Figure 5f.

### 3.2. Analysis of Phase Unwrapping

Based on the BPM, the LOS displacement field in the study area during the same period is predicted by using the BPM, and the involved BPM prediction parameters are consistent with the simulated parameters (the influence experiment of the prediction parameter errors will be carried out in the Discussion). Accordingly, the phase anti-wrapping of prediction displacement is carried out, and thus the wrapped phase and the integer cycle ambiguity of prediction displacement are calculated. Then, the wrapped phase of the residual phase is obtained by Equations (14) and (15), as shown in Figure 6a. It can be seen that the interference fringes of the wrapped phase of the residual phase are clearer and less dense, which reduces the difficulty of phase unwrapping. Based on this, the wrapped phase of the residual phase is unwrapped, and the result is shown in Figure 6b. The result shows that when using LOS displacement with the random noise and underestimation phenomenon, the unwrapped phase of the residual phase calculated by the proposed model is between −16.8 rad and 29.2 rad. Since the model and parameters of the BPM remains error-free, the unwrapped phase of the residual phase is the LOS phase error. Finally, the unwrapped phase of the residual phase is superimposed on the LOS prediction phase, and the LOS displacement can be solved by performing a transformation between phase and deformation.

### 3.3. Extracted Displacement Results and Precision Analysis

Based on the calculated LOS displacement, 3-D displacements are extracted using the proposed model, and the results are shown in Figure 7. It can be seen that the vertical displacement, strike displacement, and inclination displacement solved by the construction method are in good agreement with the real value. Since the simulated LOS displacement contains high observation noise, the smoothness of the solved 3-D displacements is poor compared with the simulated displacement fields. Following the layout method of surface deformation observation stations for coal seam mining, the observation lines are laid out above the working face along the strike and inclination main section. Furthermore, by comparing the real and calculated values of vertical displacement on the observation line L1–L2, strike displacement on the observation line L3–L4, and inclination displacement on the observation line L5–L6 (as Figure 7a,c,e), the accuracy of the solved results with the construction method is evaluated. As shown in Figure 7b,d,f, the absolute error of the vertical displacement extracted from the observation line L1–L2 is between −37.1 mm and 45.6 mm, and the root mean square error (RMSE) is 21.5 mm; the absolute error of the strike displacement extracted from the observation line L3–L4 is between −47.9 mm and 35.7 mm, with 19.0 mm of the RMSE; the absolute error of the inclination displacement extracted from the observation line L5–L6 is between −69.4 mm and 96.1 mm, with 32.9 mm of the RMSE.

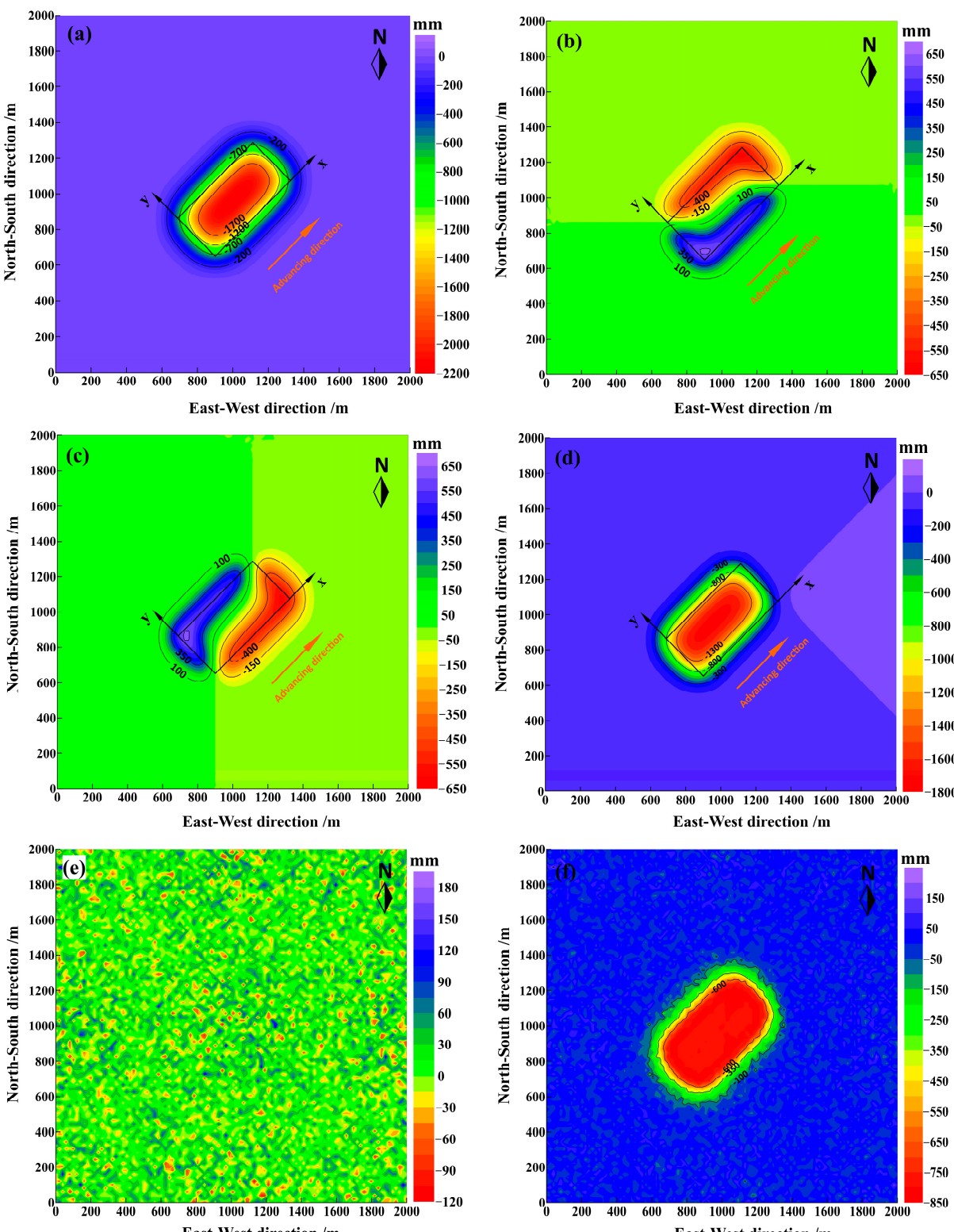

**Figure 5.** Simulated displacement fields in the mining area: (**a**) Vertical displacement field; (**b**) North displacement field; (**c**) East displacement field; (**d**) LOS displacement field; (**e**) Observation noise, and (**f**) LOS displacement field with noise and limited maximum observation.

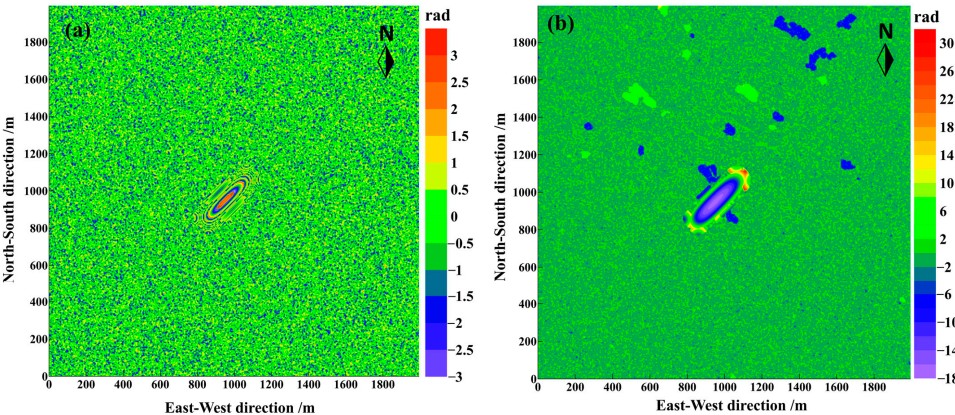

**Figure 6.** (**a**) Wrapped phase of the residual phase; (**b**) Unwrapped phase of the residual phase.

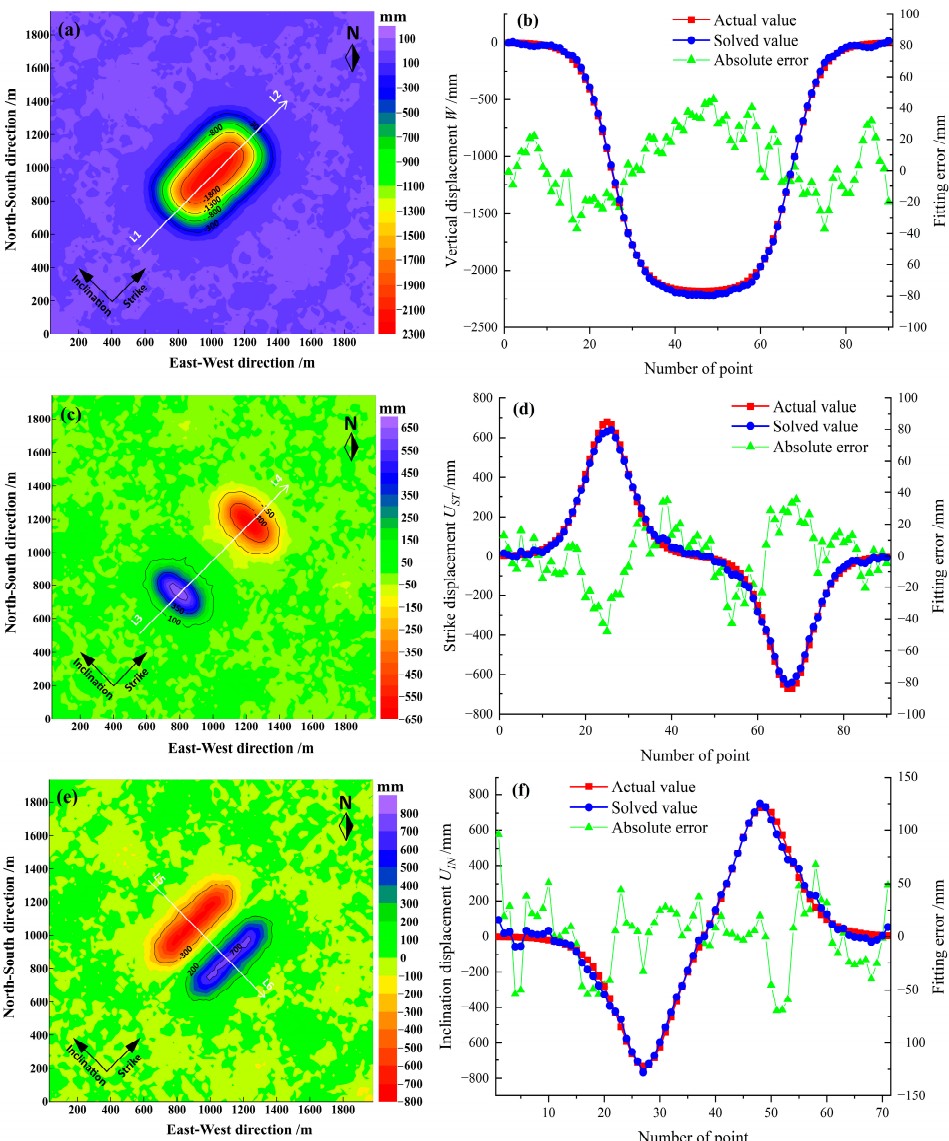

**Figure 7.** Extraction of 3-D displacement fields and accuracy evaluation by the construction method: (**a**,**b**) Extraction of the vertical displacement field, and its absolute error on the observation line L1–L2; (**c**,**d**) Extraction of the strike displacement field, and its absolute error on the observation line L3–L4; (**e**,**f**) Extraction of the inclination displacement field, and its absolute error on the observation line L5–L6.

The experimental results show that even though the LOS displacement monitored by InSAR contains large random errors, and the LOS displacement is seriously underestimated (the simulated maximum LOS observation value accounts for 46% of the real data), The solution results of the method in this paper can still have a highly consistent magnitude, distribution range, and symmetry characteristics with the simulated data. The construction method can extract the 3-D displacement field of the surface with high precision.

## 4. Real Data Experiments

### 4.1. Study Area and InSAR Data

The Huaibei mining area (marked by the red circle in the lower left panel of Figure 8) located in the northern part of Anhui Province, China, is chosen to test the proposed method. The total area of the mining area is about 9600 km$^2$, and it is an important coal production base in China. The coal mined is of good quality and belongs to "green and environment-friendly" coal. In addition to coal resources, the mining area is also rich in coalbed methane, high-quality kaolin, natural coke, and other associated mineral resources. The working face of this study is located in Shuanglong Mining of the Huaibei mining area (inside the black rectangle frame in the right panel of Figure 8), and the actual layout of the working face is shown in Figure 9. The working face started coal mining on 15 March 2021, and advanced to the stop production line (stop mining) on 15 November 2021. Geological mining conditions are as follows: the strike length is 330 m; the inclination length is 220 m; the elevation of the working face is from −179 m to −282 m; the maximum elevation difference is 103 m; the strike of the mining coal seam is near the north–south direction, near the east–west direction of the inclination; the inclined angle of the coal seam is from 6° to 18°, with an average of about 10° (near horizontal coal seam mining); the average mining height of the coal seam is about 3.3 m; the structure of the coal seam is relatively complex, including one or two layers of gangue with a thickness of 0.1–0.2 m.

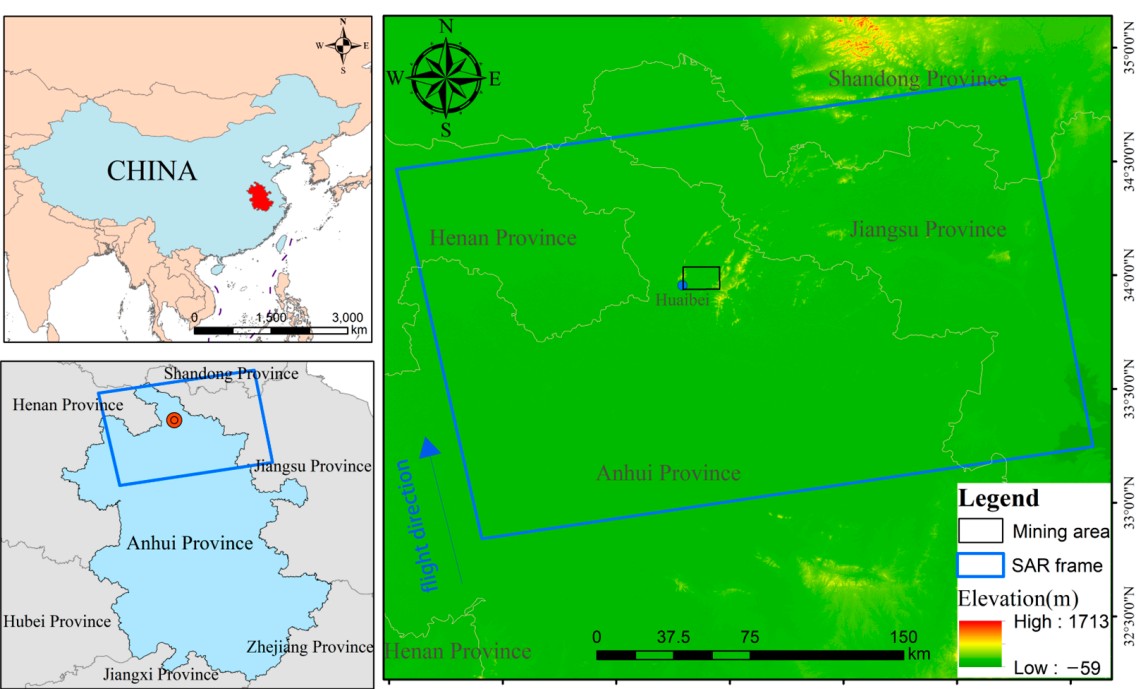

**Figure 8.** Geographic location of the Huaibei mining area (marked by the red circle in the lower left panel). The blue rectangle denotes the footprints of the collected Sentinel-1A images. The black rectangle denotes the area of Shuanglong Mining.

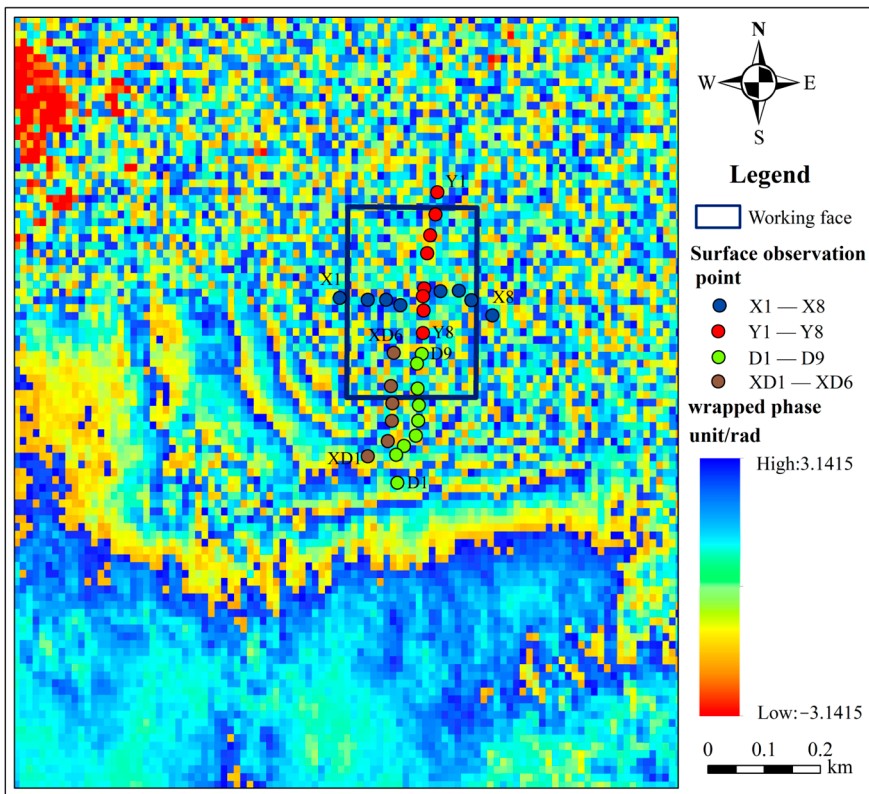

**Figure 9.** Cumulative differential interferometric phase during the period from 12 March 2021 to 10 August 2022, and layout of working face and surface observation points. The black rectangle is the mining working face. The dots are the surface displacement observation station arranged above the working face.

The radar images of the study area use Sentinel-1A data with a resolution of 5 × 20 m, and the radar frame covers the entire study area. This study uses 43 SAR images from 12 March 2021 to 10 August 2022, as the experimental data and the radar satellite monitoring period adopted is from the start of production coverage to 9 months after the stop production of the working face (parameters of the Sentinel-1A images are shown in Table 1); thus, it can be considered that the surface subsidence is basically stable. In addition, the law of rock movement and the characteristics of surface deformation in this area conform to the BPM principle. The Sentinel-1A radar satellite has a wavelength of 56 mm.

**Table 1.** Parameters of the Sentinel-1A images over the Huaibei mining area.

| Acquisition Date | Number of Scenes | Path | Frame | Incidence Angle | Heading Angle |
|---|---|---|---|---|---|
| 12 March 2021 to 10 August 2022 | 43 | 142 | 106 | 37.96° | 347.09° |

### 4.2. Displacement Phase Unwrapping in the Study Area

According to geological mining conditions, empirical formulas, and the references [38,39] related to the study area, the BPM prediction parameters of the mining area are given as $q = 0.89$, $\tan\beta = 1.76$, $b = 0.34$, $\theta = 86.5°$, $S_1 = S_2 = S_3 = S_4 = 0$ m.

On the one hand, based on the selected prediction parameters and the existing geological mining conditions, the 3-D displacement field is respectively predicted by using the BPM. Then, according to the theory of Section 2.1.2, the predicted phase of LOS displacement is solved; furthermore, the predicted wrapped phase is obtained by the phase wrapping. On the other hand, based on the obtained SAR images, the cumulative interferometric phase of the mining area during the period from 12 March 2021 to 10 August 2022, is processed

by using differential interferometric synthetic aperture radar (D-InSAR) technology. As shown in Figure 9, the closer to the center of the working face, the denser the interference fringes, and the differential interferometric phase on the side of the start production line shows regular interference fringes, while the differential interferometric phase on the side of the stop production line is chaotic and appears decoherent. The interferometric phase of the entanglement increases the difficulty of the unwrapping process. The minimum cost flow method and InSAR phase unwrapping model of the method in this paper is used for phase unwrapping, respectively. The unwrapping results of the wrapped phase are shown in Figure 10.

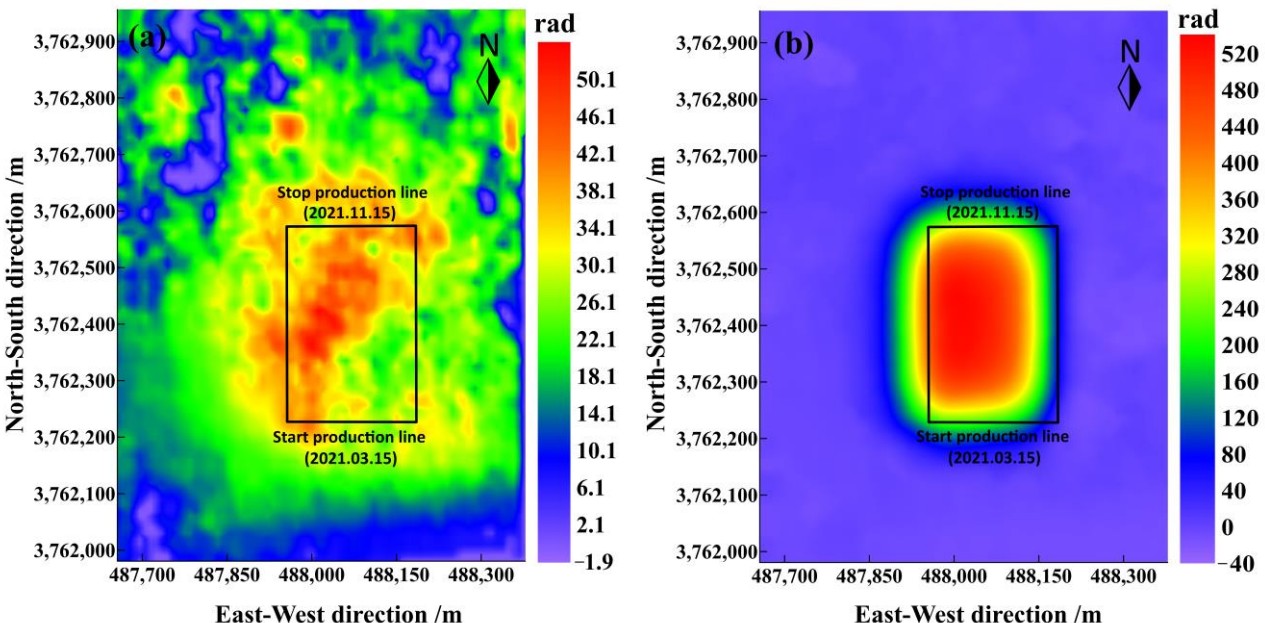

**Figure 10.** Unwrapping results of the interferometric phase in the study area: (**a**) Unwrapping results by using the minimum cost flow method; (**b**) Unwrapping results by using the proposed method.

The results show that due to the interference of surface vegetation, sedimentation ponding, and decoherent caused by large gradient deformation, the phase unwrapping results of the minimum cost flow method are relatively disordered, and the magnitude and distribution of the unwrapping phase are seriously inconsistent with the surface deformation law of mining subsidence. Moreover, the solved maximum displacement phase is only 50.14 rad. Correspondingly, the maximum deformation phase obtained by using the proposed model in this paper is 475.47 rad, and the displacement phase conforms to the law of mining subsidence. Based on the conversion principle (Equation (10)), the phase obtained by the proposed method is converted into the actual displacement, and the result is as follows in Figure 11a.

### 4.3. Extraction of Surface 3-D Displacements and Accuracy Evaluation

To solve the 3-D displacements of the surface caused by mining subsidence, on the basis of the LOS displacement of the surface obtained in Section 4.2, the 3-D displacements extraction model of coal seam mining with single-track InSAR tries to be adopted in this section, as shown in Figure 11. The calculated maximum subsidence is about 2800 mm, located near the center of the goaf. The northward displacement and eastward displacement are respectively symmetrical to the inclination main section and the strike main section. The maximum northward displacement is about 1300 mm, and the maximum eastward displacement is about 1500 mm.

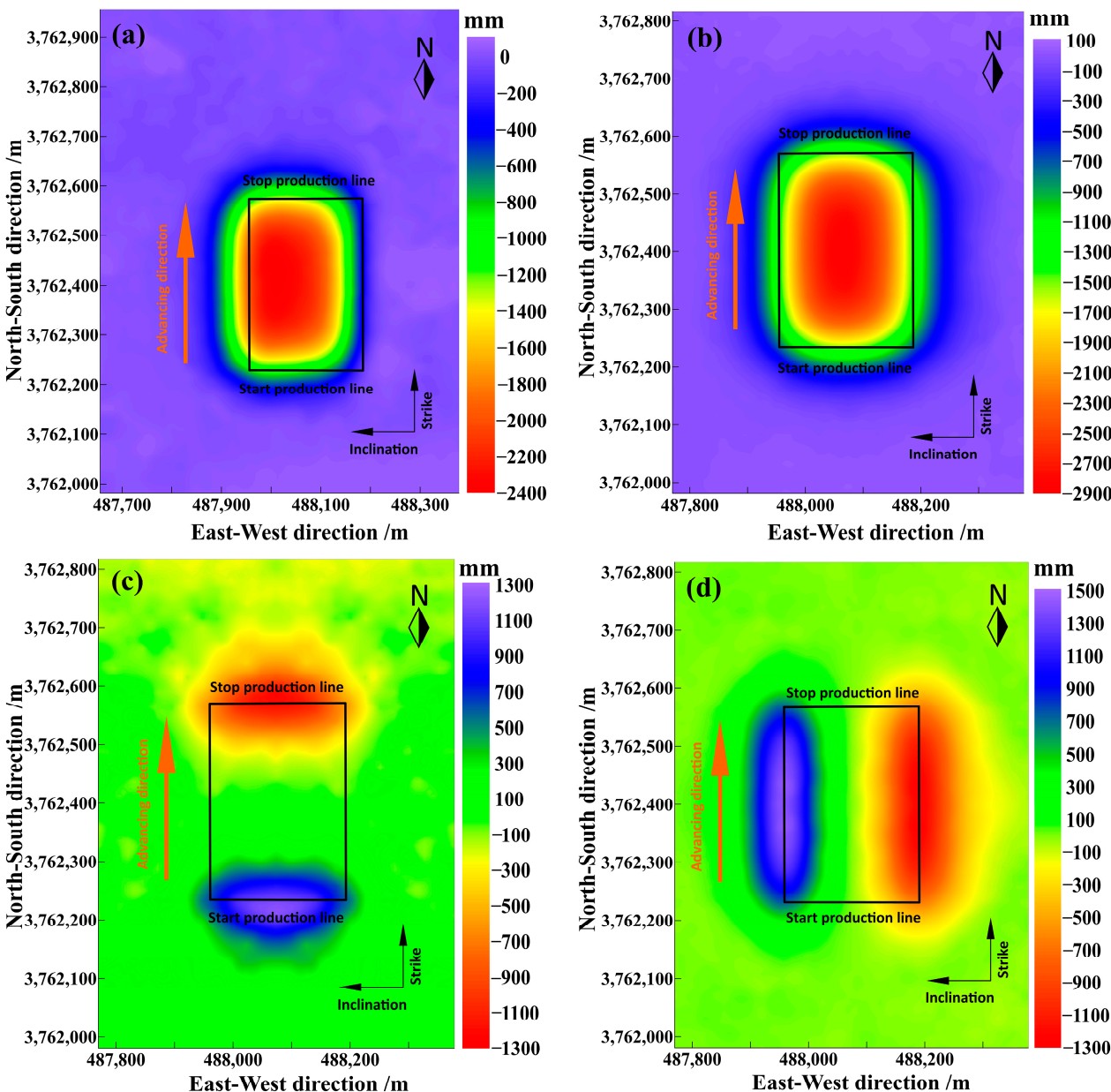

**Figure 11.** 3-D displacements extracted by the proposed method in this paper: (**a**) Solved LOS displacement in Section 4.2; (**b**) Extracted vertical displacement; (**c**) Extracted northward displacement; (**d**) Extracted eastward displacement. The black rectangle is the mining working face.

To verify the accuracy of the construction method, several surface displacement observation stations were set up on the surface above the working face during the mining process of the underground coal seam, and the monitoring period is roughly the same as that of satellite photography. A complete observation line and a half observation line are respectively laid out along the inclination main section of the working face, which mainly observe the horizontal movement along the inclination direction (that is, northward movement) and vertical displacement, and a complete observation line is laid out along the strike main section of the working face, which mainly observe the horizontal movement along the strike direction (that is, eastward movement) and vertical displacement. Hence, the 3-D displacements extracted by the method in this paper are compared with the measured values of the surface observation station, and the results are shown in Figures 12 and 13. The curves of extracted vertical displacement and horizontal displacement are in good agreement with that of measurement, and the shape of the basin is basically the same. The

fitting error of the vertical displacement is between −518.4 and 875.4 mm, with 288.2 mm of RMSE, and the fitting error of horizontal displacement is between −246.4 and 239.1 mm, with 129.5 mm of RMSE. Compared with the simulation experiment, the accuracy of the real data experiments in the mining area to solve the 3-D displacements is lower, which is caused by the following two situations through analysis: (1) The small mining depth of the working face may easily lead to discontinuity deformation such as cracks and subsidence pits on the surface; hence, the surface deformation suddenly changes in space, as shown in the dashed box in Figures 12 and 13; (2) The working face is close to the old goaf. The mining of the working face causes the activation of the old goaf; thus, the surface deformation above the working face is caused by multiple factors together.

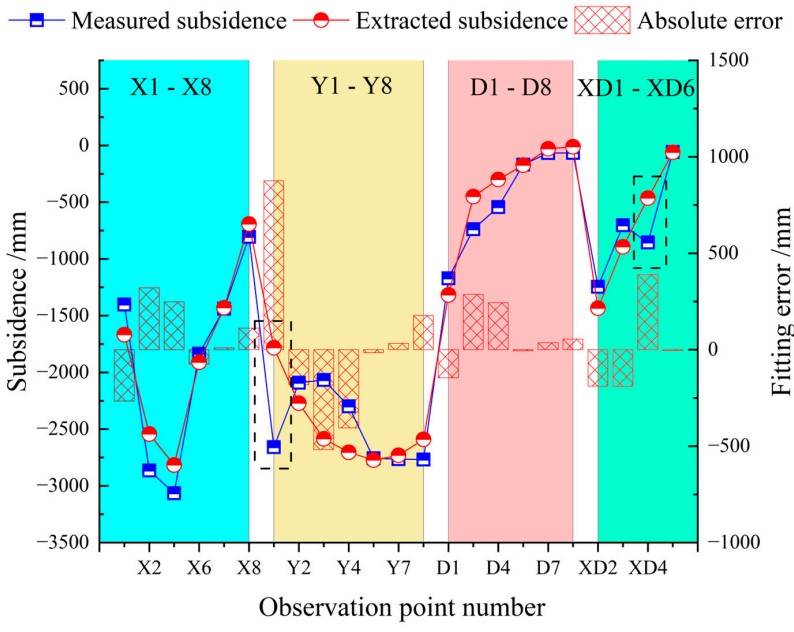

**Figure 12.** Comparison between the vertical displacement solved using the proposed method and the result of surface observation points.

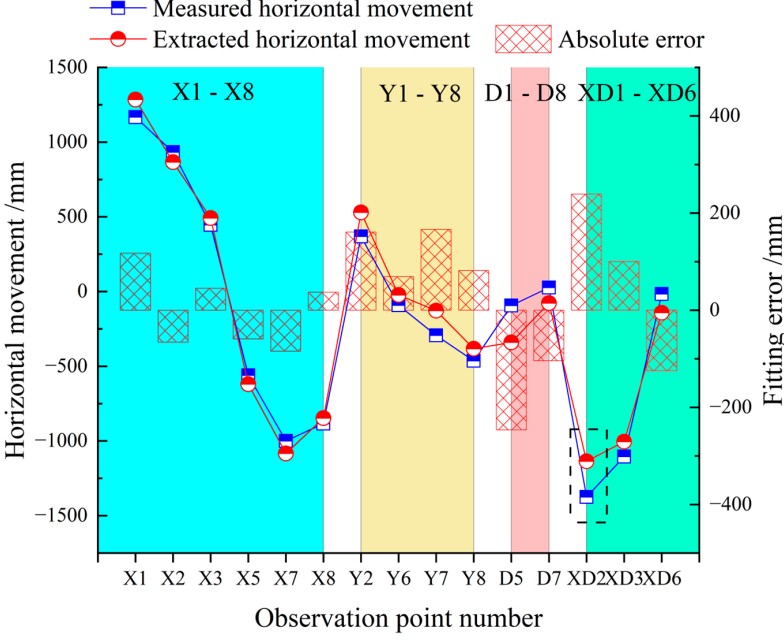

**Figure 13.** Comparison between the horizontal displacement solved using the proposed method and the result of surface observation points.

Even if the solution accuracy is slightly reduced, the RMSE of the vertical displacement accounts for only 9.4% of the maximum vertical displacement value, and the RMSE of horizontal displacement accounts for only 9.3% of the maximum horizontal displacement value. The experimental results show that the proposed method in this paper can approximate the 3-D displacements of the whole basin caused by mining subsidence, and this research method has a certain engineering application value.

## 5. Discussion

### 5.1. Influence of Various Errors on the Extraction of 3-D Displacements

In this paper, we first use the BPM to assist InSAR phase unwrapping, and the LOS displacement containing large gradient deformation is obtained. Then, based on the symmetrical characteristics of mining subsidence in a rectangular working face, the vertical displacement, strike displacement, and inclination displacement of the working face are calculated. Theoretically, the accuracy of the surface 3-D displacements extracted by using the proposed method depends on the following aspects: (1) Errors introduced by observed LOS displacement of InSAR; (2) Accuracy of the geological mining parameters $G = [m, \alpha, \varphi, H, D_3, D_1]$; (3) Accuracy of the BPM prediction parameters $P = [q, \tan\beta, b, \theta, S_1, S_2, S_3, S_4]$. For the error in aspect (1), we simulated the random error of observation and the underestimation error of observation caused by large gradient deformation in the simulation experiment, and the results show that the method in this paper can extract the 3-D displacements of the surface with better accuracy. In aspect (2), the geological mining parameters involved in the method of this paper are obtained through a precise level and traverse measurement; thus, the errors introduced are so small that they can be ignored. As far as aspect (3) is concerned, for the impact of the accuracy of the BPM prediction parameters, the following related studies are carried out based on simulated experiments.

In the simulation experiment, we added Gaussian noise to the error-free LOS simulation value and limited the maximum LOS simulation value, so as to simulate the random noise of the LOS observation and the underestimation phenomenon caused by the large gradient deformation of the mining subsidence. To test the influence of the obtained BPM prediction parameter errors on the extraction of 3-D displacements, prediction parameters with different levels of error are simulated based on the above-mentioned LOS observation error; hence, four schemes of parameter errors are set. Parameters $q$, $\tan\beta$, and $b$ are respectively added with random errors that occupy within $\pm10\%$, $\pm20\%$, $\pm30\%$, and $\pm40\%$ of themselves. Moreover, $\pm1°$, $\pm2°$, $\pm3°$, and $\pm4°$ of random errors are respectively added to the parameter $\theta$, and $\pm2$ m, $\pm4$ m, $\pm6$ m, and $\pm8$ m of random errors are respectively added to the parameter $S$. Parameters containing errors of various levels are combined into various schemes, and the parameter range of each scheme is shown in Table 2. Based on the method in this paper, the extraction experiments of 3-D displacements are carried out by using the model parameters of each scheme. To quantify the accuracy of the extraction results, the extracted values and real values on the main sections of vertical displacement, strike displacement, and inclination displacement are compared, and the RMSE on the main sections of 3-D displacements is calculated. The experimental results are shown in Figure 14.

**Table 2.** Setting scheme of the BPM prediction parameters containing different levels of error.

| Parameters | Scheme One | Scheme Two | Scheme Three | Scheme Four |
|---|---|---|---|---|
| $q$ | 0.72~0.88 | 0.64~0.96 | 0.56~1.04 | 0.48~1.12 |
| $\tan\beta$ | 1.8~2.2 | 1.6~2.4 | 1.4~2.6 | 1.2~2.8 |
| $b$ | 0.27~0.33 | 0.24~0.36 | 0.21~0.39 | 0.18~0.42 |
| $\theta$ (°) | 86~88 | 85~89 | 84~90 | 83~91 |
| $S$ (m) | −2~2 | −4~4 | −6~6 | −8~8 |

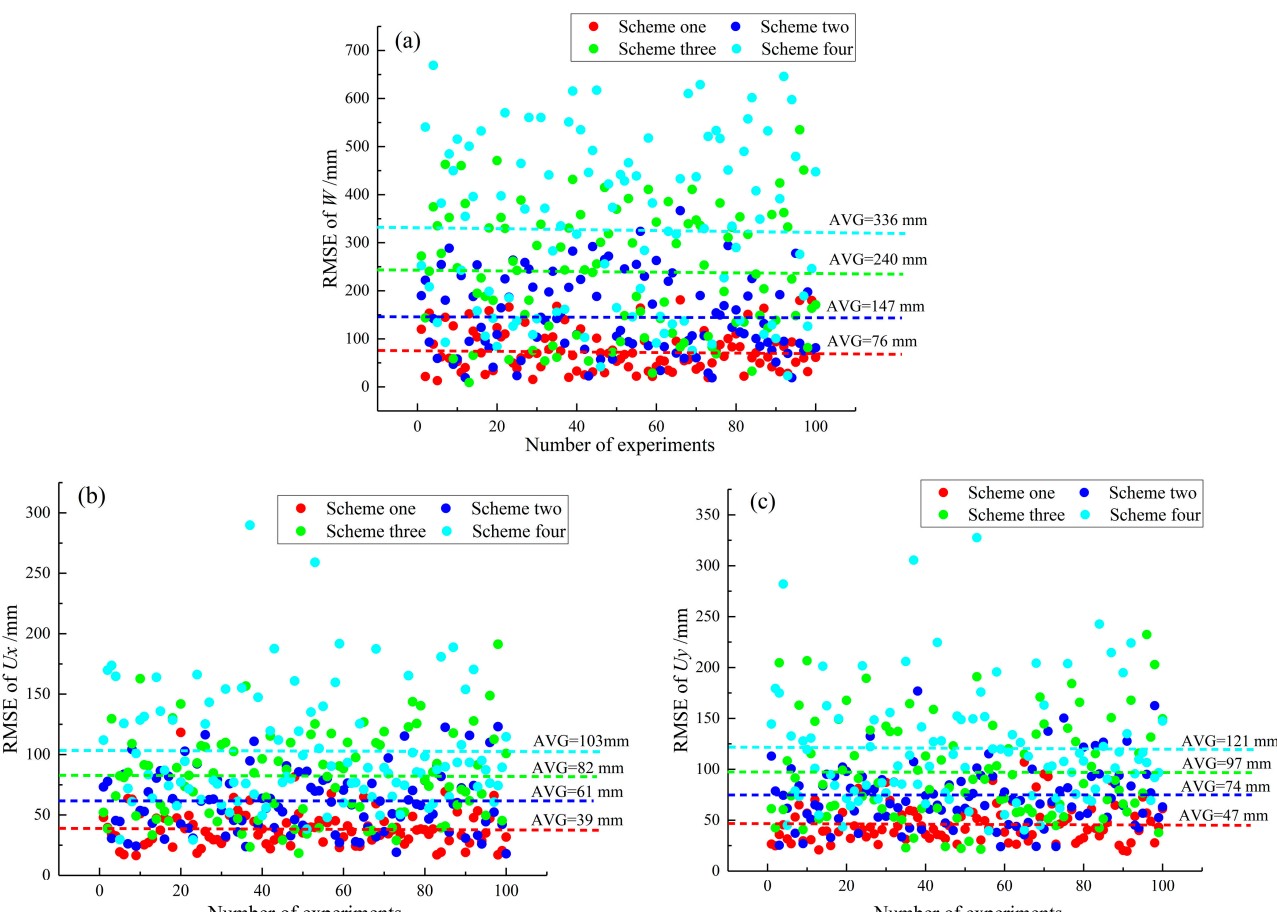

**Figure 14.** RMSE of the 3-D deformation calculated by each scheme of 100 experiments. The colored dots represent the RMSE calculated by the corresponding scheme, and the dotted line represents the average value of the RMSE with 100 experiments: (**a**) RMSE of resolved vertical displacement; (**b**) RMSE of resolved strike displacement; (**c**) RMSE of resolved inclination displacement.

Figure 14 shows that the fluctuation ranges of the RMSE of the extracted vertical displacement, strike displacement, and inclination displacement expand with the increase of the added random errors. When the parameters of Scheme One are adopted, the fluctuation ranges of the RMSE of the extracted vertical displacement, strike displacement, and inclination displacement are from 13 mm to 181 mm, from 17 mm to 118 mm, and from 20 mm to 107 mm, respectively, and the average values of the RMSE are 76 mm, 39 mm, and 47 mm, respectively. In Scheme Two, the fluctuation ranges of the RMSE of the extracted vertical displacement, strike displacement, and inclination displacement are from 19 mm to 367 mm, from 18 mm to 123 mm, and from 23 mm to 176 mm, respectively, and the average values of the RMSE are 147 mm, 61 mm, and 74 mm, respectively. In Scheme Three, the fluctuation ranges of the RMSE of the extracted vertical displacement, strike displacement, and inclination displacement are from 9 mm to 535 mm, from 18 mm to 191 mm, and from 21 mm to 232 mm, respectively, and the average values of the RMSE are 240 mm, 82 mm, and 97 mm, respectively. As far as Scheme Four is concerned, the fluctuation ranges of the RMSE of the extracted vertical displacement, strike displacement, and inclination displacement are from 23 mm to 669 mm, from 29 mm to 290 mm, and from 39 mm to 328 mm, respectively, and the average values of the RMSE are 336 mm, 103 mm, and 121 mm, respectively.

Further analysis shows that the ratio of the average RMSE of the 3-D displacements calculated by Scheme One and Scheme Two to the corresponding maximum displacement can be kept within 10% ($W$, $U_x$, and $U_y$ are 6.7%, 8.9%, and 9.9% respectively). Therefore,

in the case of considering the LOS displacement error, using the construction method in this paper, when the errors of selected model parameters $q$, $\tan\beta$, and $b$ are kept within $\pm20\%$, the error of the parameter $\theta$ is kept within $\pm2°$, and the error of the parameter $S$ is kept within $\pm4$ m, the 3-D deformation of the whole basin in the mining subsidence can be effectively monitored. It should be noted that the parameter errors in Schemes Three and Four are difficult to appear in real situations. For example, the value of parameter $q$ generally does not exceed 1, and the value of parameter $\theta$ generally does not exceed $90°$. Even if the error of selected parameters is so large, the average RMSE of the 3-D displacement extracted by this method to the corresponding maximum deformation can still be kept within 20% ($W$, $U_x$, and $U_y$ are 15.3%, 15.1, and 16.2%, respectively).

*5.2. Fusion of InSAR and Multi-Source Heterogeneous Data*

The fusion of InSAR data and multi-source heterogeneous data (such as LiDAR, 3-D laser scanner, GNSS, leveling, etc.) can realize the complementary advantages of multi-source data. Theoretically, by reasonably determining the weight of multi-source observation data and designing a feasible fusion methodology, the multi-level and high-precision 3-D displacement monitoring of the mine surface can be effectively realized. Therefore, the applicability of fusing InSAR with multi-source heterogeneous data was explored based on the proposed model in this paper.

We deduced the 3-D displacements extraction model with single-track InSAR in Section 2.2.2; following this, multi-source heterogeneous data monitored in space as one-dimensional, two-dimensional, or three-dimensional are combined. Thus, Equation (22) can be improved to Equation (23). Equation (23) takes the monitoring displacement data of 3-D laser and leveling as an example, and it needs to be further explained that the positive and negative signs of the coefficients in the matrix of Equation (23) are determined according to the quadrant where the 3-D laser point is located, while Equation (23) only characterizes the situation where the point is in the first quadrant:

$$
\begin{bmatrix}
DLOS_1 \\
DLOS_2 \\
DLOS_3 \\
DLOS_4 \\
Laser_W \\
Laser_N \\
Laser_E \\
Level_W \\
\vdots
\end{bmatrix}
=
\begin{bmatrix}
PM1 & PM2 & -PM3 \\
PM1 & PM2 & PM3 \\
PM1 & -PM2 & PM3 \\
PM1 & -PM2 & -PM3 \\
1 & 0 & 0 \\
0 & -\cos\varphi & -\sin\varphi \\
0 & -\sin\varphi & \cos\varphi \\
1 & 0 & 0 \\
\vdots & \vdots & \vdots
\end{bmatrix}
\begin{bmatrix}
W \\
U_x \\
U_y
\end{bmatrix}
\tag{23}
$$

In addition, according to the least squares criterion, $\boldsymbol{V}^{\mathrm{T}}\boldsymbol{P}_w\boldsymbol{V} = R_{\min}$, where $\boldsymbol{V}$ is the fitted residual matrix of each observation, $\boldsymbol{P}_w$ is the weight matrix of observations, and $R_{\min}$ is the minimum value under the criterion. When only single-track InSAR is used to solve 3-D displacements, $\boldsymbol{P}_w$ is taken as the identity matrix, since it is an equal-precision observation. When combining multi-source heterogeneous data, the $\boldsymbol{P}_w$ is assigned different weights due to the different precision of multi-source observation technology, so as to improve the calculation accuracy of 3-D displacement results. In this paper, the method of determining the weight is referred to in the literature [40].

To test the feasibility of the fusion method, the simulation observation data of the 3-D laser and leveling were added based on simulated experiments in Section 3, and 30 simulation observation points were selected in the study area (the positions of the simulated observation points relative to the working face are shown in Figure 15a); 3-D laser and leveling data at observation points remain error-free. Then the monitoring 3-D deformation of the 3-D laser and the monitoring vertical deformation of leveling are respectively interpolated to the same resolution of InSAR. Finally, the 3-D deformation extraction experiments fusing InSAR and leveling data, and fusing InSAR, leveling, and

3-D laser data were carried out, respectively. The calculation results of the observation line L1–L2 in Figure 15a are analyzed, and the experimental results are shown in Figure 15b–d.

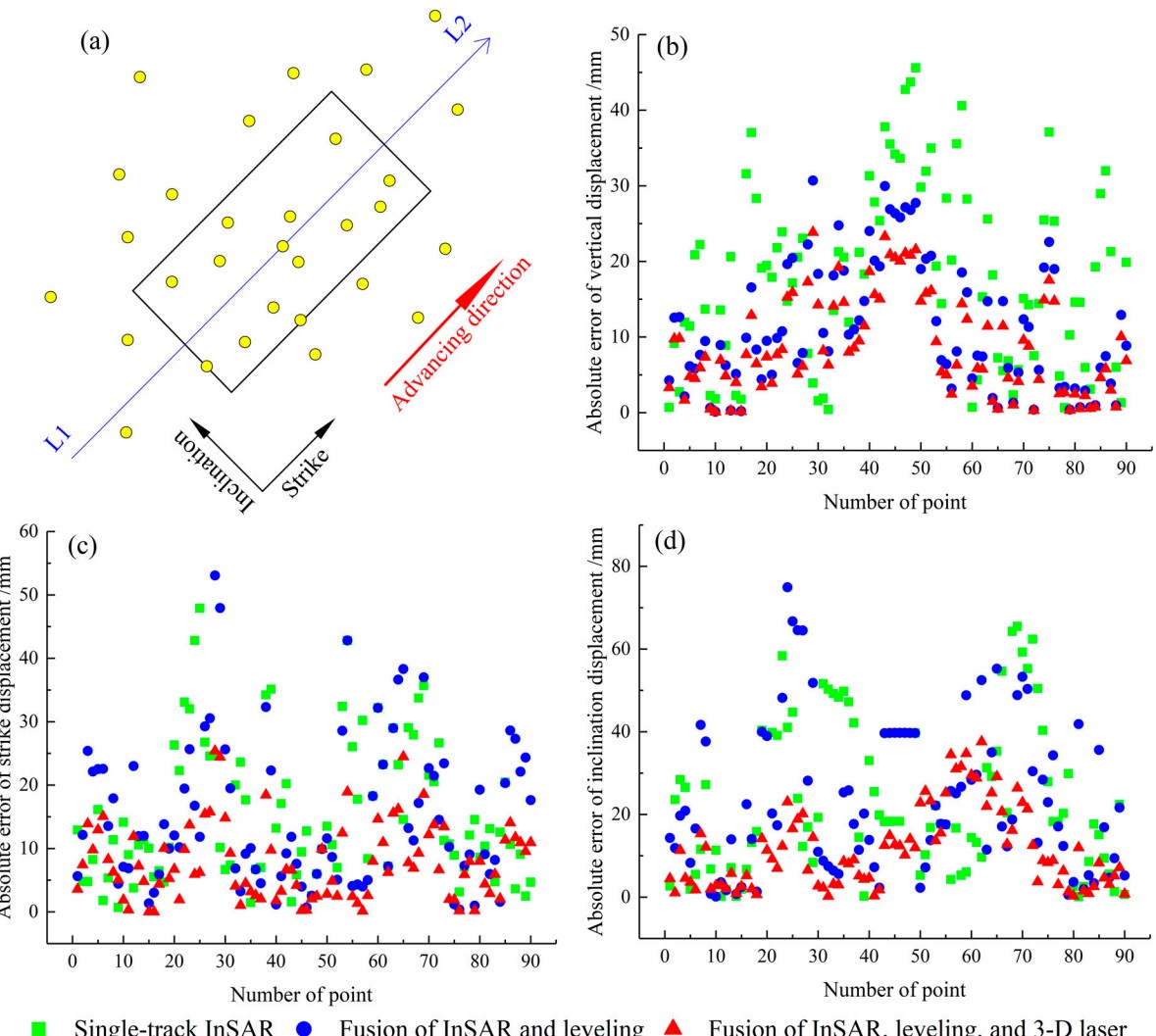

■ Single-track InSAR ● Fusion of InSAR and leveling ▲ Fusion of InSAR, leveling, and 3-D laser

**Figure 15.** (**a**) Black rectangle represents the working face, and the yellow dots represent the simulated positions of leveling and 3-D laser; (**b**–**d**) Absolute error of calculated vertical displacement, strike displacement, and inclination displacement on line L1–L2.

The calculation of the experimental results shows that the average values of the absolute errors of the vertical displacement by the single-track InSAR method, the fusion of InSAR and leveling method, and the fusion of InSAR, leveling, and 3-D laser method are 17.8 mm, 11.1 mm, and 8.6 mm, respectively. The average values of the absolute errors of the strike displacement by utilizing the three methods are 15.5 mm, 15.2 mm, and 7.8 mm, respectively. The average values of the absolute errors of the inclination displacement by utilizing the three methods are 23.6 mm, 23.5 mm, and 11.3 mm, respectively. Combined with Figure 15, it can be seen that the accuracy of extracting 3-D displacement results by the fusion of InSAR, leveling, and 3-D laser method is the best. The accuracy of vertical displacement extracted by the fusion of InSAR and leveling method is better than that of the single-track InSAR method, and inferior to the fusion of InSAR, leveling, and 3-D laser method. The accuracy of the fusion of InSAR and leveling method, and the single-track InSAR method to extract the strike displacement and inclination displacement is very close. Therefore, the fusion of InSAR and multi-source heterogeneous data can better improve the accuracy of calculated 3-D displacements in mining areas, and the relevant theories,

methods, and practical applications of fusing multi-source heterogeneous data should receive more attention.

*5.3. Extraction Model Applicable to the Mining of Inclined Coal Seam*

As mentioned above, the construction method in this paper can be better applied to the extraction of 3-D displacements in mining horizontal (or near-horizontal) coal seams. To expand the scope of application of this method, the extraction model applicable to the mining of inclined coal seam is discussed in this section. When the underground working face is inclined, as shown in Figure 3, the 3-D displacements of the symmetry points $P1$ and $P4$, $P2$ and $P3$ along the strike direction still maintain the symmetrical characteristics in magnitude and the direction; however, the maximum subsidence point is on the lower ribside in the inclination direction, and the 3-D displacements of the symmetrical points $P1$ and $P2$, $P3$ and $P4$ along the inclination direction lose their symmetrical characteristics in magnitude (that is, the values are not equal). Equation (22) is reduced from the original four known equations to two known equations; as a result, the three unknowns describing the spatial displacement cannot be solved to obtain a unique value. Therefore, whether to find the connection between the symmetry points in the inclination direction is the key issue for the wide adaptability of the model:

$$\frac{W_2}{W_1} = \frac{W^o(y_2)}{W^o(y_1)} = \frac{\left[\exp\left(-\frac{y_2-L}{R}\right) - \exp\left(-\frac{y_2}{R}\right)\right] \cdot \left[1 + \exp\left(-\frac{y_1}{R}\right)\right] \cdot \left[1 + \exp\left(-\frac{y_1-L}{R}\right)\right]}{\left[\exp\left(-\frac{y_1-L}{R}\right) - \exp\left(-\frac{y_1}{R}\right)\right] \cdot \left[1 + \exp\left(-\frac{y_2}{R}\right)\right] \cdot \left[1 + \exp\left(-\frac{y_2-L}{R}\right)\right]} = f_w \quad (24)$$

$$\frac{U_{x2}}{U_{x1}} = \frac{W^o(y_2)}{W^o(y_1)} = f_x \quad (25)$$

$$\frac{U_{y2}}{U_{y1}} = \frac{U^o(y_2)}{U^o(y_1)} = \frac{\frac{\left[(b'+\cot\theta)\cdot\exp\left(-\frac{y_2}{R}\right)+\cot\theta\right]}{\left[1+\exp\left(-\frac{y_2}{R}\right)\right]^2} - \frac{\left[(b'+\cot\theta)\cdot\exp\left(-\frac{y_2-L}{R}\right)+\cot\theta\right]}{\left[1+\exp\left(-\frac{y_2-L}{R}\right)\right]^2}}{\frac{\left[(b'+\cot\theta)\cdot\exp\left(-\frac{y_1}{R}\right)+\cot\theta\right]}{\left[1+\exp\left(-\frac{y_1}{R}\right)\right]^2} - \frac{\left[(b'+\cot\theta)\cdot\exp\left(-\frac{y_1-L}{R}\right)+\cot\theta\right]}{\left[1+\exp\left(-\frac{y_1-L}{R}\right)\right]^2}} = f_y \quad (26)$$

$$\begin{bmatrix} DLOS_1 \\ DLOS_2 \\ DLOS_3 \\ DLOS_4 \end{bmatrix} = \begin{bmatrix} PM1 & PM2 & -PM3 \\ f_w \cdot PM1 & f_x \cdot PM2 & -f_y \cdot PM3 \\ f_w \cdot PM1 & f_x \cdot PM2 & -f_y \cdot PM3 \\ PM1 & -PM2 & -PM3 \end{bmatrix} \begin{bmatrix} W \\ U_x \\ U_y \end{bmatrix} \quad (27)$$

In Section 2.1.1, the BPM for predicting surface deformation in mining areas is introduced, and we intend to use this model to establish the relationship between symmetry points in the inclination direction. Taking the symmetrical points $P1$ and $P2$ in Figure 3 as an example for analysis, the division operations on the prediction formulas of the BPM for vertical displacement, strike displacement, and inclination displacement at $P1$ and $P2$ are performed, respectively. As shown in Equations (24)–(26), the ratio relationship of the 3-D displacements between the symmetrical points is obtained. Substituting Equations (24)–(26) into Equations (21) and (22), the four known equations based on subsidence characteristics are finally restored, as Equation (27). Similarly, the 3-D displacements of space can be solved by using the least squares method. The prediction parameters of the BPM involved in the extraction model include $R$, $b'$, and $\theta$, thus the accuracy of extracting surface 3-D displacements of inclined coal seam mining largely depends on the precision of collected prediction parameters.

## 6. Conclusions

In this paper, the BPM was used to assist InSAR phase unwrapping; thus, the missing phase gradient of InSAR caused by the special monitoring environment of the mining area was restored, and higher precision LOS displacement can be calculated. In addition, the symmetric characteristics of surface subsidence in mining horizontal (or near-horizontal) coal seams were used as prior knowledge for theoretical derivation, and an extraction

method for large gradient 3-D displacements of mining areas based on single-track InSAR was constructed.

In the simulated experiments, we simulated the LOS displacement with the random noise and underestimation phenomenon caused by the large gradient deformation (the maximum value of the set LOS displacement only accounts for 46% of the actual maximum LOS displacement) as InSAR observations. The calculated results of the method in this paper can still have a highly consistent magnitude, distribution range, and symmetry characteristics with the actual data. Compared with the actual data, the RMSE of the calculated 3-D displacements on the observation line is 21.5 mm, 19.0 mm, and 32.9 mm (vertical displacement, strike displacement, and inclination displacement, respectively). Using 43 single-track Sentinel-1 images as observation data, the method in this paper was applied to the extraction of surface 3-D displacements in the Huaibei coal mine. The results show that the maximum value of vertical displacement, northward displacement, and eastward displacement is calculated as 2800 mm, 1300 mm, and 1500 mm, respectively. Comparing the calculated value with the measurement data of the surface observation station, the RMSE of vertical displacement and horizontal displacement is 288.2 mm (9.4% of the maximum vertical displacement value) and 129.5 mm (9.3% of the maximum horizontal displacement value), respectively. Moreover, it is believed that the reason for the low accuracy (compared with the simulated experiments) of the calculation is the discontinuous deformation of the ground caused by the small mining depth or the activation of the old goaf. The experimental results show that the proposed method in this paper can approximately extract the 3-D displacements of the whole basin in mining subsidence, and this research method has a certain value of engineering application.

In the discussion, the following studies were carried out: the influence of BPM parameter errors on the extraction of 3-D displacements was analyzed; the fusion model of InSAR and multi-source heterogeneous data was constructed, and the feasibility of the model was verified by experiments, and a feasible scheme for the extraction model applicable to the mining of inclined coal seam was explored. The following conclusions were obtained: (1) When the errors of selected model parameters $q$, $\tan\beta$, and $b$ are kept within $\pm20\%$, the error of the parameter $\theta$ is kept within $\pm2°$, and the error of the parameter $S$ is kept within $\pm4$ m, the 3-D displacements of the whole basin in the mining subsidence can be effectively monitored; (2) The fusion of InSAR and multi-source heterogeneous data can better improve the accuracy of calculated 3-D displacements; (3) The BPM can establish the connection between the symmetric points along the inclination direction, and the extraction equations for 3-D displacements can be reconstructed, but the error of the model parameters will be substituted into it.

**Author Contributions:** Conceptualization, K.J. and K.Y.; methodology, K.J. and K.Y.; software, K.J.; validation, K.J., K.Y. and Y.Z.; formal analysis, K.Y., K.J. and Y.L.; investigation, T.L.; resources, K.J. and K.Y.; data curation, K.Y., Y.L., T.L. and X.Z.; writing—original draft preparation, K.J. and K.Y.; writing—review and editing, K.J. and K.Y.; visualization, K.J.; supervision, K.Y., Y.Z., T.L., X.Z. and Y.L.; project administration, K.Y.; funding acquisition, K.Y. All authors have read and agreed to the published version of the manuscript.

**Funding:** This research was funded by the National Natural Science Foundation of China (41971401), the Fundamental Research Funds for the Central Universities (2022YJSDC22, 2022JCCXDC01), and the Entrusted Project of Huaibei Mining Co., Ltd. (2023-129).

**Data Availability Statement:** Not applicable.

**Acknowledgments:** The authors would like to thank the ESA/Copernicus for providing the Sentinel-1A SAR images. The authors would also like to thank the anonymous reviewers for their constructive comments and suggestions.

**Conflicts of Interest:** The authors declare no conflict of interest.

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
