# Peer review of "An Extraction Method for Large Gradient Three-Dimensional Displacements of Mining Areas Using Single-Track InSAR, Boltzmann Function, and Subsidence Characteristics"

_remotesensing, doi:10.3390/rs15112946_

Round 1

Reviewer 1 Report

General comments:

The manuscript deals with extracting large gradient 3-D displacements of mining areas in response to the limitations of surface deformation monitoring by using single-track InSAR technology. By building the BPM-assisted phase unwrapping model and the 3-D displacement extraction model based on symmetrical characteristics of mining subsidence, the authors have developed the extraction method for large gradient 3-D displacements of mining areas based on single-track InSAR. The proposed method can adapt to limited InSAR acquisitions and complex monitoring environments. In my opinion, it can be recommended for publication with a few clarifications.

 Specific comments:

1: The probability integral method (PIM) is widely used to predict mining-induced surface deformation, but the Boltzmann function prediction model (BPM) is preferred in the manuscript. Further supplementary instructions are suggested.

 2: Affected by the monitoring conditions in mining areas, the surface deformation monitored by InSAR is often missing, and whether the deformation missing point has an impact on the model of Section 2.2.

 3: Page 10, lines 348-350: The authors claim that “By analyzing the SAR satellite monitoring capability, time and space resolution, and the advancing speed of the working face, the maximum cumulative deformation of InSAR monitoring is set at 800 mm”. Is there relevant literature to support this approach?

 4: In Simulated experiments, the final calculation results are vertical displacement, strike displacement, and inclination displacement, while the calculation results of Real data experiments are vertical displacement, east-west displacement, and north-south displacement. Why are there two different results?

Reviewer 2 Report

Dear authors,

This manuscript proposes a strategy that uses a simulated prediction model to compensate/assist InSAR phase unwrapping for LOS results conduction. The method retrieves the 3-D displacements of mining with a 3-D deformation model by considering the relationship between InSAR-LOS and 3-D deformation. The proposed method can solve practical application problems since in mining areas, using InSAR to obtain deformation is often challenging due to the large gradient of deformation and heavy vegetation, resulting in very little available InSAR data. The structure of this manuscript is well-organized, and the authors have conducted solid work. The presented model is verified by simulation experiments, and the 3-D displacement retrieval strategy is validated with real experiments and ground truth. The performance and effectiveness are discussed with simulation data. However, one main concern is that the paper highlights the prediction of deformation to assist InSAR phase unwrapping, which should be further clarified or verified. So, the following comments are provided point-by-point, which may enhance the quality of this manuscript..

Comments:

1. The title "An extraction method" is too vague to reflect your advanced method. It would be better to consider some words related to the characteristics of your method/strategy/model.

2. Regarding the keywords, do you think "symmetric characteristics" is the highlight of your work?

3. In the introduction, the third paragraph should directly state that three categories have been developed and then introduce them one-by-one (I, II, III). In the current format, it is unclear for the reader to follow up.

4. In Line-109, could you please give some clarification or some citations to support the statement "the model has a better fit at the boundary of the deformation basin."

5. In Section 2.1.1, it would be better to give a construction figure of mining to illustrate all the geometric parameters or the parameters like (strike direction, working face, inclination direction), similar to Figure 2.

6. In line-287, what is the meaning of "multi-view"? Does it mean a stack of SAR datasets?

7. In section 3, the authors use simulation experiments to validate the performance of the proposed model. However, in my opinion, it only validates the effectiveness of the result from the prediction model and the output of the estimation model. It seems that the authors are missing the most important part, which is that the prediction model is just an assistant for the phase unwrapping. It would be much better to add some tests in this major part, such as adding some phase unwrapping operations during the simulation experiments and discussing the accuracy. Moreover, the authors can also add a small section in the real experiments/discussion part to compare/validate the results in some areas with mild deformation. If the unwrapped phase is concise with the predicted results, it would be more convincing for the readers.

8. It would be better to clarify the advantages of the proposed method over existing strategies or models, like the references in [18-20].

In my opinion, the paper is well-written, but there are some areas that could benefit from careful revision.

Reviewer 3 Report

The paper proposes a method to recover 3-D ground deformations components for the study of mining regions. The key idea, also overexploited in several other research studies in the recent past, is to use a model for the unwrapped phases derived from some physical laws to reduce the fringe rates and then have the possibility to improve significantly the retrieval accuracy of the LOS components. Using a single track to derive the parameters of such a model, of course, allows the authors to determine the global 3-D evolution of ground displacements. The method is definitely correct, however, the authors must focus more and more on the appropriateness of the used model and less on the implications related to phase unwrapping. If the model is correct, the unwrapped phases are better retrieved but this, as is, is not new and does not deserve a publication. Instead, what could deserve a publication is a profound study/understanding of the potential and limitations of the adopted model and any other potential implication for the future. For instance, in case that several tracks could be available, the determination of the model parameters could be better. Is there a potential for the application of such a strategy to a multi-platform/viewing situation? More insights on the real new issues must be included in the revised paper.

Round 2

Reviewer 2 Report

Dear authors,

Many thanks for your effort to address my comments. I have no more comments on it.

Reviewer 3 Report

The revised paper is sufficient enough for publication.